# New Evidence for *Artemisia absinthium* as an Alternative to Classical Antibiotics: Chemical Analysis of Phenolic Compounds, Screening for Antimicrobial Activity

**DOI:** 10.3390/ijms241512044

**Published:** 2023-07-27

**Authors:** Zhihao Liu, Xiaolin Li, Yan Jin, Tiegui Nan, Yuyang Zhao, Luqi Huang, Yuan Yuan

**Affiliations:** State Key Laboratory for Quality Ensurance and Sustainable Use of Dao-di Herbs, National Resource Center for Chinese Materia Medica, China Academy of Chinese Medical Sciences, Beijing 100700, China; lzh_zykh@163.com (Z.L.);

**Keywords:** *Artemisia absinthium*, antimicrobial activity, bacterial resistance, compound identification

## Abstract

*Artemisia absinthium*, an important herb of the *Artemisia* genus, was evaluated in this study for its potential as an alternative to classical antibiotics. The antimicrobial activity of methanol extracts of *A. absinthium* (MEAA) was evaluated using the broth microdilution method, revealing that *A. absinthium* exhibited broad-spectrum antibacterial and antifungal activity. Ultra-performance liquid chromatography-quadrupole-time of flight mass spectrometry (UPLC-Q-TOF-MS) was used to analyze the chemical profile of the MEAA, with a focus on flavonoids, quinic acids, and glucaric acids. A total of 90 compounds were identified, 69 of which were described for the first time in *A. absinthium*. Additionally, a new class of caffeoyl methyl glucaric acids was identified. The main active compounds were quantified and screened for antimicrobial activity. *A. absinthium* was found to be rich in quinic acids and flavonoids. The screening for antimicrobial activity also revealed that salicylic acid, caffeic acid, casticin, and 3,4-dicaffeoylquinic acid had varying degrees of antimicrobial activity. The acute toxicity of MEAA was examined following OECD guidelines. The administration of 5000 mg/kg bw of MEAA did not result in mortality in male and female mice. Furthermore, there were no observed effects on the visceral organs or general behavior of the mice, demonstrating the good safety of MEAA. This study provides new evidence for the use of *A. absinthium* as an alternative to classical antibiotics in addressing the problem of bacterial resistance.

## 1. Introduction

The emergence and spread of drug-resistant strains pose a serious threat to global public health [1]. To address the declining susceptibility of strains to antibiotics, researchers have proposed alternative therapies, such as traditional herbal medicines, probiotics, vaccines, immunoglobulins, and bacteriophages. These therapies can help reduce selection pressure and minimize resistance to classical antibiotics [2,3,4,5,6]. Traditional herbal medicines have been used for thousands of years to treat bacterial infections, with *Artemisia* species in particular excelling in traditional antimicrobial herbal medicine [7,8]. Extracts from *Artemisia* species, such as *Artemisia annua*, *Artemisia argyi,* and *Artemisia indica,* have demonstrated varying degrees of antimicrobial activity against different strains [9,10,11]. Thus, it is worth examining the antimicrobial activity of another important species in the *Artemisia* genus, *A. absinthium*.

*A. absinthium*, a perennial herb in the Asteraceae family, is widely distributed in Iran, India, Pakistan, and European countries. This herb has a long history of medicinal use, particularly in Europe and Central Asia, where it is used to treat fever, stomachache, indigestion, anorexia, and hepatitis. *A. absinthium* is rich in essential oils, flavonoids, phenolic acids and terpenoids [12] (Figure 1), and modern pharmacological studies have shown that it possesses antimicrobial, antioxidant and neuroprotective pharmacological activities. Notably, *A. absinthium* is widely used not only in medicine but also in food, cosmetics, and animal husbandry. In the food sector, *A. absinthium* is an important compound in absinthe, which has been consumed in Europe for more than 300 years and is believed to have appetizing, stomachic, and tonic properties. In cosmetics, *A. absinthium* is used in a range of products based on its excellent antimicrobial and antioxidant activity. In the livestock industry, *A. absinthium* is used as a special feed additive to reduce the risk of intestinal diseases in livestock [13,14]. Given the numerous practical applications of *A. absinthium*, it is suggested that it possesses superior antimicrobial activity. Although there are many reports analyzing the antimicrobial activity of the essential oil of *A. absinthium*, there is still a lack of research on the composition analysis and evaluation of the antimicrobial activity of the methanol extracts of *A. absinthium* (MEAA).

In recent years, advanced technologies such as high-resolution mass spectrometry (HRMS) [15,16], ion mobility spectrometry [17], diagnostic product ions (DPIs) filters [18], and liquid chromatograph mass spectrometer (LC-MS) data acquisition [19] have been continuously developed and improved, enabling metabolomics studies based on high-performance liquid chromatography-tandem mass spectrometry to be widely used in the metabolite analysis and structural characterization of traditional herbal medicines. However, the identification of positional and geometric isomers among a large number of metabolites, as well as the identification of new chemical structures, is often hindered by the lack of standards and reference data [20,21]. To address this issue, previous studies have analyzed and summarized various metabolite identification strategies based on limited representative standards, such as MS/MS fragmentation laws [22], ion mobility spectra [17], and quantitative structure-retaining relationships [23]. In addition, chromatographic retention time prediction based on molecular hydrogen bonding analysis and Clog*P* calculation for isomers has also been used for isomer identification [24]. In this study, we analyzed and summarized the identification strategy based on past studies to characterize the chemical profile of *A. absinthium*.

In this study, we aimed to investigate the potential of *A. absinthium* as an alternative to classical antibiotics by evaluating the antimicrobial activity of its methanol extracts. Additionally, we conducted an analysis of the MEAA using ultra-performance liquid chromatography-quadrupole-time of flight mass spectrometry (UPLC-Q-TOF-MS) and employed structure analysis and isomer identification strategies to focus on flavonoids, quinic acid, and glucaric acid compounds. We also quantified the main active compounds present in *A. absinthium* using ultra-performance liquid chromatography-tandem mass spectrometry (UPLC-MS/MS) and evaluated their antimicrobial activity through the broth microdilution method. By identifying the compounds and conducting quantitative analysis, antimicrobial activity screening, and evaluation of acute toxicity, we aimed to provide more evidence for *A. absinthium* as a viable alternative to classical antibiotics.

## 2. Results and Discussion

### 2.1. Antimicrobial Activity of MEAA

This study investigated the antimicrobial activity of MEAA against nine clinically common strains, including four Gram-positive and four Gram-negative bacteria, and one fungal strain. The broth microdilution method was used to determine the MIC and MBC values of the methanol extracts. The results, presented in Table 1, demonstrate that the MEAA exhibits broad-spectrum antibacterial activity with different degrees of potency against the tested strains. Gram-positive bacteria were found to be more susceptible to the methanol extracts compared to Gram-negative bacteria, as indicated by lower MIC/MBC values. Notably, the methanol extracts demonstrated superior activity against *S. aureus* (MIC = 1.25 mg/mL, MBC = 1.25 mg/mL), *S. epidermidis* (MIC = 0.625 mg/mL, MBC = 1.25 mg/mL), *P. aeruginosa*, and *B. cereus* (MIC = 1.25 mg/mL, MBC = 1.25 mg/mL), with *S. epidermidis* being the most sensitive among the tested bacteria. The fungus *C. albicans* was also inhibited by the methanol extracts (MIC = 2.5 mg/mL, MBC = 5.0 mg/mL).

Similar to some other *Artemisia* species, such as *A. annua*, *A. argyi*, and *A. indica*, *A. absinthium* also exhibited antimicrobial activity, particularly against Gram-positive bacteria. This is consistent with what has been reported so far. This may be due to differences in the structure of cell membranes between Gram-positive and Gram-negative bacteria. Hydrophilic compounds, which are present in the methanol extracts, require permeation across the membrane to be effective. However, the presence of lipopolysaccharides in the outer membrane of Gram-negative bacteria can limit the penetration of hydrophobic antibacterial compounds, resulting in the need for higher doses of these compounds [25,26].

### 2.2. Chemical Profile of A. absinthium

The chemical profile of *A. absinthium* was investigated using Ultra Performance Liquid Chromatography-Quadrupole-Time of Flight Mass Spectrometry (UPLC-Q-TOF-MS) (Figure 2). The identified compounds were compared with reference standards and literature data, and a total of 90 compounds were tentatively identified, including 31 flavonoids, 20 quinic acids, 17 glucaric acids, 8 other organic acids, and 14 other polyphenols. Among these, 69 compounds were identified for the first time in *A. absinthium*. Detailed information about these identified compounds is presented in Table 2. This study used a comprehensive and relatively complete identification strategy, integrating the identification strategies of previous researchers, to effectively identify many positional and geometric isomers, thereby solving the problem of compound identification in *A. absinthium*.

#### 2.2.1. Flavonoids

Flavonoids are a class of compounds with a basic parent nucleus consisting of 2-phenylchromogenic ketones that are widely present in plants. They usually combine with sugar residue to form flavonoid glycosides, while a small proportion exists in the form of free flavonoid aglycone [27]. In this study, we identified six flavonoid aglycones, namely kaempferol (#62), apigenin (#84), chrysoeriol (#86), eupatilin (#87), casticin (#88), and artemetin (#90) [13]. Notably, eupatilin was identified for the first time in *A. absinthium*. Additionally, we identified 25 flavonoid glycosides, which can be divided into six categories based on their different flavonoid aglycones. These six subtypes of flavonoids can be distinguished by their specific diagnostic ions, which are as follows: 285.0399/284.0321 for kaempferol-, 269.0450/268.0372 for apigenin-, 301.0348/300.0270 for quercetin-, 345.0610/344.0532 for Spinacetin-, 315.0505/314.0427 for isorhamnetin-, 331.0454/330.0376 for mearnsetin-type flavonoids [20]. However, determining the positional isomers for flavonoids is challenging. Previous studies have shown that the positional isomers for flavonoids can be identified based on their glycosylation sites, types of aglycone, and the multiple diagnostic product ions (DPIs) and DPIs ratio of compounds [28]. To illustrate the identification strategy for such flavonoid glycosides, we take kaempferol-type flavonoids as an example. For kaempferol *O*-monoglycosides, the peak intensity ratio (PR) of the kaempferol radical (*m*/*z* 284.03) to the deprotonated kaempferol ion (*m*/*z* 285.04) was below 1 for the 7-*O*-glycosides, but the PR (>1) was higher for 3-*O*-glycosides (Figure 3B). This can be used to distinguish the positional isomers of 3-*O*-glycosides and 7-*O*-glycosides. Therefore, we identified kaempferol 7-glucoside (#65) [20]. For 3-*O*-diglycosides and 7-*O*-diglycosides, under negative ionization, the moieties linkage at the 7-*O* position is preferentially cleaved to the 3-*O* position, thus the relative abundance of [M-H-7-*O* glycoside]^−^ is greater than [M-H-3-*O* glycoside]^−^ (Figure 3C). Moreover, the DPIs and DPIs ratio of these compounds were affected by the different linking modes of glycoside chains and the types of sugar [20,29]. Based on the above identification strategy, we identified six kaempferol *O*-diglycosides (#51, #53, #59, #60, #62, #68). The identification strategy described above is applicable to other types of flavonoid glycosides as well. As a result, we were able to identify four spinacetin-type flavonoids (#50, #55, #57, #80), three isorhamnetin-type flavonoids (#72, #76, #82), two mearnsetin-type flavonoids ((#63, #66), three apigenin-type flavonoids (#69, #71, #74), and three quercetin-type flavonoids (#57, #61, #64).

#### 2.2.2. Quinic Acids

The diagnostic ion for quinic acids is usually *m*/*z* 191.06. In this study, 20 quinic acids were identified (Figure 4A), which are formed by the combination of quinic acid with caffeoyl-, coumaroyl-, and feruloyl- [30]. Among them, eight compounds(#19, #21, #27, #30, #32, #36, #37, #40) with [M-H]^−^ ions at *m*/*z* 353.0873 were identified as caffeoylquinic acids (Figure 4B). The diagnostic product ions of caffeoylquinic acids are 707.1823[2M-H]^−^, 353.0873 [M-H]^−^, 191.0556 [M-H-caffeoyl-]^−^, and 179.0344 [M-H-quinic]^−^ according to the fragmentation rule (Figure 4C). The eight isomers are formed by the combination of *cis*-caffeoyl- and *trans*-caffeoyl- on the OH groups at the 1,2,3,4 positions of quinic acid [31]. The elution order of positional isomers is determined by Clog*P*, with smaller Clog*P* indicating a smaller retention time. After calculating the Clog*P* of positional isomers and combining it with the literature data, the elution order of 4 positional isomers is 3-/5-/4-/1-caffeoylquinic acids (Figure 4D) [20,32]. To further identify the geometric isomers, molecular simulation results summarized the formation rules of intramolecular hydrogen bonds of these isomers. It showed that 5-*cis*-caffeoylquinic acids were more hydrophobic and eluted later than 5-*trans*-caffeoylquinic acids, while 3-*cis*- and 4-*cis*-caffeoylquinic acids were the opposite. The possible reason for this phenomenon is that 5-*cis*- are able to form at least one hydrogen bond that is not present in 5-*trans*-caffeoylquinic acids, and that hydrogen bonds cannot be formed in 3-*cis*- and 4-*cis*-caffeoylquinic acids [32]. Therefore, the structures of 4 pairs of caffeoylquinic acids isomers were tentatively identified, with the elution order of isomers being 3-*cis*, 3-*trans*, 5-*trans*, 5-*cis*, 4-*cis*, 4-*trans*, 1-*cis*, and 1-*trans* [20]. The PR of DPI at *m*/*z* 191.06 to *m*/*z* 179.03 showed remarkably consistent change among these isomers. The *trans*-isomers had higher PR values than their corresponding *cis*-isomers. Among the eight caffeoylquinic acid isomers, 4-*trans*-caffeoylquinic acid had the highest PR value, reaching 113.70. Additionally, the PR values of 1-*cis*-caffeoylquinic acid were similar to those of 3-*cis*-caffeoylquinic acid, while the PR values of 4-*cis*-caffeoylquinic acid were similar to those of 5-*cis*-caffeoylquinic acid. Similarly, the PR values of 1-*trans*-caffeoylquinic acid were similar to those of 5-*trans*-caffeoylquinic acid. The DPI ratios and elution order based on ClogP can accurately identify positional and geometric caffeoylquinic acids or other acylated quinic acids. Based on the PRs of DPI at *m*/*z* 191.06 to *m*/*z* 163.04, 3 p-coumaroylquinic acids were identified [31]. Additionally, four dicaffeoylquinic acids (#67, #70, #73, #79), one tricaffeoylquinic acid (#83), and three feruloylquinic acids (#48, #54, #58) were identified, referring to the literature data [33,34,35].

#### 2.2.3. Glucaric Acids

Glucaric acids are a group of compounds found widely in plants, formed by the combination of glucaric acid and one to four acylated or combined residues [29]. These compounds have a unique structure of positional and geometric isomers that remain the focus of ongoing research [36]. In this study, we identified eight compounds (#8, #9, #10, #12, #15, #17, #20, #22) with [M-H]^−^ ions at *m*/*z* 371.0614 as caffeoylglucaric acids (Figure 5A). The diagnostic product ions of caffeoylglucaric acids are 743.1307 [2M-H]^−^, 371.0614 [M-H]^−^, 209.0297 [M-H-caffeoyl-]^−^, 191.0192 [M-H-caffeoyl-H_2_O]^−^ and 179.0344 [M-H-glucaric]^−^ (Figure 5B) [30]. To further analyze the positional and geometric isomers, we compared the intra-molecular hydrogen bonds of these isomers. The intra-molecular hydrogen bonds formed by 1(6)-OH and 2(5)-CO in 2-*trans*- and 5-*trans*-caffeoylglucaric acids were found to be more stable than in 3-*trans* and 4-*trans* caffeoylglucaric acids [37]. Therefore, the 3-*trans* and 4-*trans* caffeoylglucaric acids eluted with a shorter retention time than the 2-*trans*- and 5-*trans*-caffeoylglucaric acids. Additionally, because the intra-molecular hydrogen bond on the same side is more stable than the intra-molecular hydrogen bond on the reverse side, the total intra-molecular hydrogen bonds of 3-*trans*-caffeoylglucaric acid formed at 3-/4-and 3-/2- were weaker than that of 4-*trans*-caffeoylglucaric acid. Likewise, the 5-*trans*-caffeoylglucaric acid was stronger than its 2-trans isomer (2-*trans*-caffeoylglucaric acid). Therefore, the elution order of 4 *trans*-isomers was 3-*trans*, 4-*trans*, 2-*trans*, and 5-*trans* [32]. Based on the DPIs ratio rules of previous studies [37,38], the elution order of 4 pairs of caffeoylglucaric acid isomers was 3-*trans*, 4-*trans*, 3-*cis*, 4-*cis*, 2-*trans,* 5-*trans*, 2-*cis*, and 5-*cis*. After calculating the PRs of DPI at *m*/*z* 209.03 to *m*/*z* 191.02, the developed PR rules were supplemented (Figure 5C). The PR values of *cis*-caffeoylglucaric acid were similar to that of the corresponding *trans*-caffeoylglucaric acid, which was consistent with previous studies [39]. Furthermore, the PRs of 3-*cis*, 4-*cis*- and 5-*cis*-caffeoylglucaric acid were higher than that of corresponding 3-*trans*, 4-*trans*- and 5-*trans*-caffeoylglucaric acid, and the PR values of 2-*trans*-caffeoylglucaric acid were higher to that of the corresponding 2-*cis* caffeoylglucaric acid. Therefore, the PR rules supplemented on the basis of previous studies could be used to accurately identify the caffeoylglucaric acid isomers and other acylated caffeoylglucaric acids.

**Table 2 ijms-24-12044-t002:** All identified compounds in *A. absinthium* using the summarized identification strategy based on UPLC-Q-TOF-MS.

No.	RT	Formula	[M-H]^−^ Measured	(*m*/*z*) Predicted	Δ (ppm)	ESI-MS/MS (*m*/*z*)	Identification	Ref.
1	0.55	C_6_H_12_O_7_	195.0496	195.0505	−4.6	195.0496, 177.0385, 129.0175	Gluconic acid	[38]
2	0.59	C_7_H_12_O_6_	191.0545	191.0556	−5.8	383.1183 ([2M-H]^−^), 191.0545, 179.0544, 173.0440, 127.0384, 85.0280	Quinic acid	[32]
3	0.62	C_4_H_6_O_5_	133.0128	133.0137	−6.8	133.0128, 115.0018, 85.0276, 71.0127	Malic acid	[13]
4	0.71	C_4_H_6_O_5_	133.0126	133.0137	−9.8	133.0128, 115.0018, 85.0276, 71.0127	L-(−)-malic acid	[13]
5	0.87	C_6_H_8_O_7_	191.0184	191.0192	−4.2	383.0455 ([2M-H]^−^), 191.0181, 111.0070, 87.0071	Citric acid	[13]
6	0.93	C_6_H_8_O_7_	191.0179	191.0192	−6.8	383.0465 ([2M-H]^−^), 191.0179, 111.0073, 87.0075	Isocitric acid	[13]
7	1.47	C_7_H_6_O_5_	169.0124	169.0137	−7.7	169.0124, 125.0241	3,4,5-Trihydroxybenzoic acid	[34]
8	1.74	C_15_H_16_O_11_	371.0605	371.0614	−2.7	743.1323 ([2M-H]^−^), 371.0605, 209.0285, 191.0181, 135.0430, 85.0277	3-*trans*-Caffeoylglucaric acid	[29,30]
9	1.93	C_15_H_16_O_11_	371.0600	371.0614	2.4	743.1322 ([2M-H]^−^), 371.0623, 209.0275, 191.0188, 135.048, 85.0283	4-*trans*-Caffeoylglucaric acid	[29,30]
10	2.06	C_15_H_16_O_11_	371.0604	371.0614	−2.4	743.1312 ([2M-H]^−^), 371.0605, 209.0285, 191.0179, 135.0426, 85.0281	3-*cis*-Caffeoylglucaric acid	[29,30]
11	2.18	C_13_H_16_O_9_	315.0708	315.0716	−1.3	631.1515 ([2M-H]^−^), 315.0708, 153.0171, 152.0098, 109.0270, 108.0199	Protocatechuic acid 3-glucoside	[13]
12	2.27	C_15_H_16_O_11_	371.0607	371.0614	0.5	743.1342 ([2M-H]^−^), 371.0616, 209.0293, 191.0185, 135.0437, 85.0281	4-*cis*-Caffeoylglucaric acid	[29,30]
13	2.46	C_14_H_18_O_9_	329.0872	329.0871	−0.6	659.1819 ([2M-H]^−^), 329.0872, 167.0332	Vanillic acid glucoside	[13]
14	2.52	C_7_H_6_O_4_	153.0177	153.0188	−7.2	153.0180, 109.0268	Protocatechuic acid	[40]
15	2.58	C_15_H_16_O_11_	371.0603	371.0614	0.3	743.1326 ([2M-H]^−^), 371.0615, 209.0287, 191.0181, 135.0436, 85.0280	2-trans-Caffeoylglucaric acid	[29,30]
16	2.67	C_13_H_16_O_9_	315.0715	315.0716	−1.0	315.0713, 153.0175, 152.0112, 109.0275, 108.0197	Protocatechuic acid 4-glucoside	[13]
17	2.76	C_15_H_16_O_11_	371.0611	371.0614	−0.8	743.1360 ([2M-H]^−^), 371.0615, 209.0288, 191.0195, 135.0440, 85.0278	5*-trans*-Caffeoylglucaric acid	[29,30]
18	3.00	C_15_H_20_O_10_	359.0974	359.0978	−1.1	719.2036 ([2M-H]^−^), 359.0976, 197.0440, 153.539, 135.0437	Glucosyringic acid	[13]
19 +	3.04	C_16_H_18_O_9_	353.0867	353.0873	−1.4	353.0867, 191.0542, 179.0335, 135.0441	3-*cis*-Caffeoylquinic acid	[31]
20	3.08	C_15_H_16_O_11_	371.0610	371.0614	−0.3	743.1329 ([2M-H]^−^), 371.0605, 209.0290, 191.0181, 85.0281	2-*cis*-Caffeoylglucaric acid	[29,30]
21	3.20	C_16_H_18_O_10_	353.0853	353.0873	3.7	707.1833 ([2M-H]^−^), 353.0886, 191.0542, 179.0327, 173.0440, 135.0439, 85.0284	3-*trans*-Caffeoylquinic acid	[31]
22	3.28	C_15_H_16_O_11_	371.0609	371.0614	−1.3	371.0609, 209.0285, 191.0547, 135.0437, 85.0276	5-*cis*-Caffeoylglucaric acid	[29,30]
23 *	3.31	C_16_H_18_O_11_	385.0757	385.0771	−3.6	385.0757, 223.0437, 191.0174	3-*trans*-Caffeoyl methyl glucaric acid	[37]
24	3.35	C_7_H_12_O_5_	175.0596	175.0606	−5.7	175.0596, 157.0489, 113.0591	3-Isopropylmalic acid	[34]
25 +	3.45	C_7_H_6_O_3_	137.0227	137.0239	−4.4	137.0233, 93.0321	Salicylic acid	[34]
26	3.50	C_15_H_16_O_9_	339.0705	339.0716	−3.2	339.0707, 177.0173, 133.0274	Esculin	[34]
27 +	3.57	C_16_H_18_O_11_	353.0867	353.0873	1.0	707.1833 ([2M-H]^−^), 353.0857, 191.0544, 179.0328, 173.0432, 135.0433, 85.0277	5*-trans*-Caffeoylquinic acid	[31]
28 *	3.59	C_16_H_18_O_11_	385.0761	385.0771	−2.6	385.0761, 223.0433, 191.0182	4-*trans*-Caffeoyl methyl glucaric acid	[37]
29	3.60	C_7_H_12_O_5_	175.0595	175.0606	−6.3	175.0590, 157.0486, 113.0590	2-Isopropylmalic acid	[13]
30	3.74	C_16_H_18_O_12_	353.0868	353.0873	1.4	707.1833 ([2M-H]^−^), 353.0857, 191.0540, 179.0330, 173.0433, 135.0436, 85.0283	5-*cis*-Caffeoylquinic acid	[31]
31	3.86	C_8_H_14_O_5_	189.0753	189.0763	−5.3	189.0755, 127.0742, 99.0799	3-Hydroxyoctanedioic acid	[13]
32	3.99	C_16_H_18_O_13_	353.0862	353.0873	1.1	707.1831 ([2M-H]^−^), 353.0846, 191.0556, 179.0338, 173.0437, 135.0437, 85.0282	4-*cis*-Caffeoylquinic acid	[31]
33 *	4.04	C_16_H_18_O_11_	385.0761	385.0771	−2.6	385.0761, 223.0440, 191.0173	3-*cis*-Caffeoyl methyl glucaric acid	[37]
34 +	4.07	C_18_H_16_O_8_	359.0760	359.0767	−1.9	359.0760, 197.0435, 179.0332, 161.0224	Rosmarinic acid	[34]
35	4.14	C_8_H_14_O_5_	189.0752	189.0763	−5.8	189.0752, 127.0742, 99.0803	2-Hydroxyoctanedioic acid	[13]
36 +	4.20	C_16_H_18_O_14_	353.0864	353.0873	−0.6	707.1838 ([2M-H]^−^), 353.0876, 191.0549, 179.0335, 173.0439, 135.0436, 85.0280	4-*trans*-Caffeoylquinic acid	[31]
37	4.38	C_16_H_18_O_15_	353.0862	353.0873	−1.1	707.1815 ([2M-H]^−^), 353.0857, 191.0542, 179.0328, 173.0438, 135.0437, 85.0277	1-*cis*-Caffeoylquinic acid	[31]
38 *	4.39	C_16_H_18_O_11_	385.0753	385.0771	−0.5	771.1613 ([2M-H]^−^), 385.0769, 223.0441, 205.0325, 191.0175	2-*trans*-Caffeoyl methyl glucaric acid	[37]
39 *	4.45	C_16_H_18_O_11_	385.0767	385.0771	−1.0	771.1616 ([2M-H]^−^), 385.0767, 223.0443, 205.0339, 191.0181	4-*cis*-Caffeoyl methyl glucaric acid	[37]
40 +	4.56	C_9_H_8_O_4_	179.0334	179.0344	−5.6	179.0334, 135.0441	Caffeic acid	[39]
41 *	4.68	C_16_H_18_O_11_	385.0761	385.0771	−2.6	385.0761, 223.0443, 205.0333, 191.0183	2-*cis*-Caffeoyl methyl glucaric acid	[37]
42	4.74	C_18_H_24_O_12_	431.1188	431.1190	−0.5	863.2485 ([2M-H]^−^), 431.1188,305.0687, 269.1235, 225.1118, 125.0226	Asperulosidic acid	[34]
43	4.97	C_15_H_16_O_9_	339.0710	339.0716	−2.7	339.071, 307.0846, 209.0291, 161.0232	Daphnin	[13]
44	5.02	C_16_H_18_O_17_	353.0863	353.0873	−2.8	707.1839 ([2M-H]^−^), 353.0863, 191.0545, 179.0333, 173.0436, 135.0438, 85.0280	1-*trans*-Caffeoylquinic acid	[31]
45 *	5.05	C_16_H_18_O_11_	385.0774	385.0771	0.8	385.0774, 223.0438, 191.0186	5-*cis*-Caffeoyl methyl glucaric acid	[37]
46	5.08	C_18_H_28_O_9_	387.1642	387.1655	−3.4	387.1642, 207.1004	7-Epi-12-Hydroxyjasmonic acid glucoside	[13]
47	5.17	C_18_H_28_O_9_	387.1657	387.1655	0.5	387.1657, 255.062, 207.1013	Tuberonic acid glucoside	[13]
48	5.27	C_16_H_18_O_8_	337.0913	337.0923	−3.0	675.1931 ([2M-H]^−^), 337.0913, 191.0543, 179.0343, 173.0450, 163.0386	3-*p*-Coumaroylquinic acid	[32]
49 *	5.61	C_16_H_18_O_11_	385.0765	385.0771	−1.6	385.0765, 223.0438, 191.0175	5-*trans*-Caffeoyl methyl glucaric acid	[37]
50	5.71	C_29_H_34_O_18_	669.1664	669.1667	−0.4	669.1664, 507.1128, 345.0580, 344.0476, 330.0353, 329.0276, 301.0327, 300.0263, 285.0025	Spinacetin 3-O-glucosyl-(1->6)-glucoside	[20]
51	5.82	C_32_H_38_O_20_	741.1891	741.1878	1.8	741.1891, 580.1392, 579.1348, 285.0388, 284.0316	Kaempferol 3-sambubioside-7-glucoside	[20]
52	5.88	C_17_H_20_O_9_	367.1023	367.1029	−1.6	367.1023, 193.0494, 191.0545, 173.0443, 135.0429, 134.0357, 93.0329	3-Feruloylquinic acid	[29]
53	5.96	C_33_H_40_O_20_	755.2041	755.2035	0.8	755.2041, 593.1494, 285.0388, 284.0320	Kaempferol 7-caffeoylrhamnosyl-glucoside	[29]
54	6.05	C_16_H_18_O_8_	337.0922	337.0923	−0.3	675.1938 ([2M-H]^−^), 337.0922, 191.0548, 179.0343, 173.0442, 163.0397	5-*p*-Coumaroylquinic acid	[32]
55	6.25	C_17_H_20_O_9_	367.1034	367.1029	1.4	367.1034, 193.0496, 191.0545, 173.0456, 135.0435, 134.0360, 93.0334	5-Feruloylquinic acid	[29]
56	6.54	C_17_H_20_O_9_	367.1028	367.1029	−0.3	367.1028, 193.0488, 191.0541, 173.0442, 135.0431, 134.0345, 93.0322	4-Feruloylquinic acid	[29]
57	6.83	C_21_H_20_O_12_	463.0879	463.0877	0.4	463.0879, 301.0331, 300.0267, 271.0231, 255.0271, 243.0290	Quercetin 7-glucoside	[20]
58	6.86	C_16_H_18_O_8_	337.0913	337.0923	−3.0	675.1944 ([2M-H]^−^), 337.0913, 191.0542, 179.0323, 173.0433, 163.0367	4-*p*-Coumaroylquinic acid	[32]
59	7.04	C_26_H_28_O_15_	579.1357	579.1350	1.2	579.1357, 413.1424, 285.0388, 284.0305	Kaempferol 7-xylosylglucoside	[28]
60	7.11	C_26_H_28_O_15_	579.1346	579.1350	−0.7	579.1346, 461.0720, 285.0387, 284.0313	Kaempferol 7-sambubioside	[28]
61 +	7.13	C_27_H_30_O_16_	609.1456	609.1456	0.0	609.1456, 301.0328, 300.0262, 271.0235, 255.0284, 243.0285	Rutin	[26]
62	7.31	C_27_H_30_O_15_	593.1504	593.1506	−0.3	593.1504, 447.0935, 285.0388, 284.0311	Kaempferol 3-rhamnoside-7-galactoside	[20]
63	7.36	C_28_H_32_O_17_	639.1560	639.1561	−0.2	639.1560, 331.0426, 330.0359, 315.0125, 287.0179, 271.0226, 243.0279	Mearnsetin 3-rutinoside	[28]
64 +	7.38	C_21_H_20_O_12_	463.0881	463.0877	0.9	463.0881, 301.0330, 300.0269, 271.0236, 255.0285, 243.0287	Isoquercitrin	[26]
65	7.50	C_21_H_20_O_11_	447.0927	447.0927	0.0	447.0927, 402.1311, 285.0385, 284.0312, 227.0316	Kaempferol 7-glucoside	[20]
66	7.62	C_22_H_22_O_13_	439.0984	439.0982	0.2	493.0984, 331.0429, 330.0367, 315.0314, 287.0186	Mearnsetin 3-glucoside	[28]
67 +	7.90	C_25_H_24_O_12_	515.1187	515.1190	0.2	1031.2423 ([2M-H]^−^), 515.1182, 353.0863, 191.0542, 179.0332, 135.0432	3,4-Dicaffeoylquinic acid	[33]
68	7.99	C_27_H_30_O_15_	593.1510	593.1506	0.7	593.1510, 447.0916, 285.0388, 284.0310, 255.0287, 227.0333	Kaempferol 7-rutinoside	[20]
69	8.06	C_26_H_28_O_14_	563.1387	563.1401	−2.5	563.1401, 269.0439, 268.0363	Apigenin-7-apioglucoside	[20]
70	8.2	C_25_H_24_O_12_	515.1195	515.1190	0.4	1031.2491 ([2M-H]^−^), 515.1182, 353.0863, 191.0542, 179.0332, 135.0432	3,5-Dicaffeoylquinic acid	[33]
71	8.25	C_27_H_30_O_14_	577.1557	577.1557	0	577.1557, 271.0231, 269.0437, 268.0366	Apigenin 7-neohesperidoside	[20]
72	8.27	C_28_H_32_O_16_	623.1632	623.1647	−2.4	623.1632, 315.0502, 314.0428, 300.0285, 299.0190, 272.0292, 271.0237, 255.0287, 243.0287	Isorhamnetin 7-rutinoside	[20]
73	8.37	C_25_H_24_O_12_	515.1193	515.1190	0.6	1031.2510 ([2M-H]^−^), 515.1189, 353.0860, 191.0545, 179.0335, 135.0437	1,3-Dicaffeoylquinic acid	[33]
74	8.40	C_27_H_28_O_14_	575.1410	575.1401	1.6	575.1410, 431.1005, 413.0874, 269.0442, 268.0300	Apigenin 7-[6″-(3-hydroxy-3-methylglutaryl)glucoside]	[29]
75	8.62	C_29_H_34_O_17_	653.1725	653.1718	1.1	653.1731, 345.0607, 344.0529, 330.0362, 329.0295, 315.0114, 314.0060, 302.0411, 301.0345, 287.0181, 286.0107, 259.0255, 258.0158	Spinacetin 7-rutinoside	[20]
76	8.67	C_22_H_22_O_12_	477.1039	477.1033	1.3	477.1039, 315.0486, 314.0418, 300.0252, 299.0188, 272.0273, 271.0237, 258.0156, 243.0282, 242.0204, 215.0331	Isorhamnetin 3-glucoside	[20]
77	8.82	C_29_H_34_O_17_	653.1730	653.1718	1.8	653.1730, 345.0605, 344.0525, 330.0350, 329.0289, 287.0191, 271.0245, 258.0155	Spinacetin 7-robinobioside	[20]
78	9.00	C_18_H_30_O_8_	373.1862	373.1862	0	373.1862, 211.1326, 193.1219	6-Epi-7-isocucurbic acid glucoside	[20]
79 +	9.01	C_25_H_24_O_12_	515.1195	515.1190	1.0	1031.2501 ([2M-H]^−^), 515.1182/353.0863/191.0542/179.0333/135.0432	4,5-Dicaffeoylquinic acid	[33]
80	9.09	C_23_H_24_O_13_	507.1150	507.1139	2.2	507.1150, 345.0602, 344.0530, 330.0358, 329.0293, 302.0395, 301.0346, 287.0168, 286.0107	Spinacetin-3-glucosid	[28]
81	9.46	C_22_H_22_O_11_	461.1082	461.1084	−0.7	461.1082, 299.0536, 298.0471, 255.0282	Kaempferide 3-glucoside	[20]
82	10.22	C_24_H_24_O_13_	519.1152	519.1139	2.5	519.1152, 315.0502, 314.0076, 300.0272, 299.0198, 271.0237, 258.0151	Isorhamnetin 7-(2″-acetylglucoside)	[20]
83	12.27	C_34_H_30_O_15_	677.1506	677.1497	2.5	677.1497, 515.1186, 353.0826, 191.0544, 179.0338, 161.0225, 135.0428	3,4,5-Tricaffeoylquinic acid	[31]
84 +	13.47	C_15_H_10_O_5_	269.0442	269.0450	−3.0	269.0442, 117.0365	Apigenin	[34]
85 +	13.73	C_15_H_10_O_6_	285.0395	285.0399	−1.4	285.0395, 255.0284, 227.0349	Kaempferol	[13]
86	14.16	C_16_H_12_O_6_	299.0544	299.0556	−2.7	299.0544, 241.0953, 215.0549	Chrysoeriol	[26]
87	17.93	C_18_H_16_O_7_	343.0823	343.0818	1.5	343.0818, 328.0570, 313.0321, 298.0078	Eupatilin	[26]
88 +	18.50	C_19_H_18_O_8_	373.0919	373.0923	−0.5	373.0921, 358.0681, 343.0445, 257.0088, 229.0125	Casticin	[34]
89 +	18.64	C_16_H_12_O_6_	299.0547	299.0556	−2.7	299.0547, 285.0353, 284.0323, 257.1288	Kaempferide	[34]
90	19.01	C_20_H_20_O_8_	387.1074	387.1080	−1.5	387.1074, 356.9952	Artemetin	[34]

Note: *, unknown compounds; +, compounds identified using chemical reference compounds.

#### 2.2.4. A New Class of Caffeoyl Methyl Glucaric Acid Isomers

During the process of identifying feruloylglucaric acids, we identified eight compounds (#23, #28, #33, #38, #39, #41, #45, #49) with an *m*/*z* of 385.0711 [29]. However, the diagnostic product ions of these compounds (385.0711, 223.0454, 205.0348) were significantly different from those of feruloylglucaric acids (385.0711, 209.0297, 191.0192) (Figure 6A). In the case of feruloylglucaric acids, the DPI at 209.0297 was produced by the removal of a feruloyl- from feruloylglucaric acids, while the DPI at 223.0454 was produced by the removal of a caffeoyl- from the newly discovered compounds. Feruloyl- has one more methyl- than caffeoyl-. In the case of [M-H]^−^ ions consistency, those compounds that we identified might be formed by the combination of caffeoyl- and a methylated glucaric acid. After checking databases such as PubChem (https://pubchem.ncbi.nlm.nih.gov/, accessed on 30 January 2023) and ChemSpider (http://www.chemspider.com/, accessed on 30 January 2023), the DPI at 223.0454 might be derived from methyl glucaric acid. Therefore, we tentatively identified these compounds as a new class of caffeoyl methyl glucaric acid isomers (Figure 6B). To identify positional and geometric isomers, we followed the identification rules of glucaric acid isomers. Hence, the elution order of caffeoyl methyl glucaric acids was tentatively identified as 3-*cis*, 4-*cis*, 3-*trans*, 2-*cis*, 4-*trans*, 2-*trans*, 5-*trans*, and 5-*cis*.

#### 2.2.5. Other Compounds

In accordance with previous studies [33,34,35,40,41,42], the analysis revealed the presence of 5 phenolic acids, namely 3,4,5-trihydroxybenzoic acid (#7), protocatechuic acid (#14), salicylic acid (#25), rosmarinic acid (#34), and caffeic acid (#40). Moreover, eight other organic acids (#3, #4, #5, #6, #24, #29, #31, #35) were identified. In addition, eight phenolic glycosides (#11, #13, #16, #18, #42, #46, #47, #78) were identified from *A. absinthium*.

### 2.3. Content of 14 Active Compounds in A. absinthium

The MEAA was found to be rich in phenolic acids and flavonoids, which are known to have antimicrobial properties and are commonly used in traditional herbal medicine [43,44]. For instance, the casticin, chlorogenic acid, 3,5-dicaffeoylquinic acid, 4,5-dicaffeoylquinic acid, and cynaroside are the quality control compounds of the traditional antimicrobial Chinese medicine *Vitex trifolia* fruit and *Lonicera japonica* flower in the Chinese Pharmacopoeia [45]. To more accurately determine the content of antimicrobial compounds in *A. absinthium* and to establish its quality standards, we selected eight phenolic acids and six flavonoids and determined their content. Among these compounds, chlorogenic acid, rutin, kaempferide, and casticin had higher content than others, with values up to 25.5950 ± 0.544 μg/g, 11.3730 ± 0.368 μg/g, 8.8500 ± 0.258 μg/g, and 3.5470 ± 0.148 μg/g, respectively. The results of the content determination of the 14 compounds are shown in Table 3.

### 2.4. Antimicrobial Activity of 14 Active Compounds in A. absinthium

To further investigate the antimicrobial effect of MEAA, we examined the antimicrobial activity of 14 active compounds against nine strains. After the determination of MIC and MBC values of different compounds against different strains, the results revealed that salicylic acid, caffeic acid, casticin, and 3,4-dicaffeoylquinic acid had strong antimicrobial effects and inhibited all nine strains to varying degrees. However, the remaining 10 compounds had higher MBC values than 1 mg/mL for different strains and did not exhibit significant sensitivity. Salicylic acid displayed a good inhibitory effect on all nine strains, as indicated by its low MIC and MBC values. The sensitivity of Gram-positive bacteria to the four compounds was higher than that of Gram-negative bacteria, consistent with the results of the methanol extracts. Among the fungal strains, only salicylic acid and caffeic acid exhibited potent antimicrobial activity against *C. albicans*. Detailed data are shown in Table 4.

The four compounds screened were more effective in antibacterial activity compared to the other compounds, which seems to explain the antibacterial activity of the methanol extracts. Salicylic acid, caffeic acid, and 3,4-dicaffeoylquinic acid belong to the class of phenolic acids, which are known for their proven antimicrobial activity [45]. Salicylic acid and caffeic acid have been extensively studied for their antimicrobial effects, which involve reducing the hardness of bacterial cell walls and breaking down bacterial cell membranes [46]. Among the quinic acids, 3,4-dicaffeoylquinic acid is more sensitive to nine strains, and further investigation is required to understand the differences in the antimicrobial activity of dicaffeoylquinic acid isomers due to their structural specificity. Fiamegos et al. [33] reported that dicaffeoylquinic acid isomers act as a pump inhibitor that targets the efflux pump system of Gram-positive bacteria, revealing a potential mechanism for bacterial inhibition by dicaffeoylquinic acid isomers. The sensitivity of dicaffeoylquinic acid isomers to Gram-positive bacteria is consistent with the finding of the current study. The study also highlights the potential antimicrobial activity of casticin, the primary active compound in *Vitex trifolia* fruit, which is known for its antitumor and anti-inflammatory activity [46]. Additionally, further trials are needed to confirm whether the newly identified caffeoyl methyl glucaric acids have antimicrobial activity.

### 2.5. Acute Toxicity

During the observation period, mice in both the control and treated groups exhibited no mortality. Activity levels, hair condition, and defecation did not reveal any significant abnormalities in any of the groups. After oral administration of MEAA, body weight changes remained within the normal physiological range for all groups of mice for 14 days (Figure 7). No significant differences were found in body weight between male and female mice in the treated group when compared to the control group (Table 5). The major organs, including the heart, spleen, kidney, lung, liver, and thymus, were examined in each group of mice. No significant lesions were observed in any of the major organs in all groups of mice (Figure A1). The organ coefficients of male and female mice in both control and treated groups are presented in Table 6. There were no significant differences in the organ coefficients of mice in the treated group compared to those in the control group.

The above results indicate that the LD_50_ of MEAA is greater than 5000 mg/kg bw, demonstrating that MEAA possesses a good safety profile. This is consistent with the reported claims of better safety of other *Artemisia* plant extracts. Mekonen et al. reported on the acute toxicity of the aqueous extracts of *Artemisia afra*, which did not cause mortality in mice at a dose of 5000 mg/kg bw, indicating a good safety profile [47]. Similarly, Dib’s study showed that after a single dose of 6000 mg/kg bw of crude extract of *Artemisia campestris* L., no significant abnormalities were observed in mice after 14 days [48]. In addition, there are still many reports that support the better safety of *Artemisia* plant extracts. Many pieces of evidence show that a multitude of *Artemisia* plant extracts exert therapeutic effects with better safety, which also applies to MEAA.

## 3. Materials and Methods

### 3.1. Plant Materials and Chemicals

The dried plants of *A. absinthium* were obtained from Xinjiang herbal medicine market in China and were confirmed as genuine using a molecular identification method that we had previously established [49]. The HPLC-grade solvents (methanol and acetonitrile) were purchased from Merck KGaA (Darmstadt, Germany). Formic acid was purchased from Fisher Scientific (Geel, Belgium). Chemical reference compounds such as salicylic acid (LOT:20110801, ≥98%), caffeic acid (LOT:20060809, ≥98%), rosmarinic acid (LOT:20081109, ≥98%), chlorogenic acid (LOT:20032007, ≥98%), cryptochlorogenic acid (LOT:21042003, ≥98%), neochlorogenic acid (LOT:21030108, ≥98%), 3,4-dicaffeoylquinic acid (LOT:21072216, ≥98%), 4,5-dicaffeoylquinic acid (LOT:20072216, ≥98%), kaempferol (LOT:20071503, ≥98%), apigenin (LOT:20040403, ≥98%), casticin (LOT:21050613, ≥98%), rutin (LOT:20050610, ≥98%), isoquercitrin (LOT:21041613, ≥98%), and kaempferide (LOT:21912010, ≥98%) were purchased from Beijing Beit Renkang Bio-medical Technology Co. (Beijing, China). Each chemical reference compound was prepared in 70% methanol as a 1 g/L master batch and diluted into a gradient concentration of 0.001, 0.01, 0.1, 0.5, 1, 10 mg/L for quantitative analysis.

### 3.2. Preparation of Plant Extracts

The Dried plants were pulverized into a fine powder using a tissue crusher. A sample of 100 mg of the powder was placed into a centrifuge tube, to which 2 mL of 70% methanol was added. The mixture was extracted using sonication at 25 °C and 40 KHz for 30 min. Following centrifugation at 13,000 rpm for 10 min, the supernatant was filtered through a 0.2 μm membrane filter and utilized for chemical analysis. The analytical concentration was 50 mg/mL (*w*/*v*). Another 100 g of the sample powder was extracted using the same extraction method as described above to obtain 2 L of extract, which was allowed to settle and subsequently filtered. The extract was then filtered, concentrated to a paste using a rotary evaporator, and dried in a freeze dryer for use in the screening of antimicrobial activity and the evaluation of acute toxicity.

### 3.3. Experimental Animals

For the acute toxicity study, 20 SPF-graded ICR mice, consisting of 10 females and 10 males with an average weight of 18–22 g, were obtained from Beijing Vital River Laboratory Animal Technology Co., Ltd. (Beijing, China), with an animal license no. SYXK (Beijing, China) 2021-0006. The mice were housed in the SPF-grade animal facility at the Institute of Traditional Chinese Medicine, Chinese Academy of Traditional Chinese Medicine, under standard laboratory conditions (12 h light/dark cycle, 23 ± 2 °C). They were fed on a standard rodent diet and could drink freely. These experiments were conducted following internationally accepted guidelines for evaluating the safety of MEAA [50] and were approved by the Animal Welfare Ethics Committee of the Institute of Traditional Chinese Medicine, Chinese Academy of Traditional Chinese Medicine.

### 3.4. Chemical Profile Analysis by UPLC-Q-TOF-MS

A chemical profile analysis of *A. absinthium* was conducted on a Waters Xevo G2-S Q-TOF-MS (Waters, Milford, MA, USA) with Masslynx V 4.1 software. The separation of compounds was performed using a Waters Acquity UPLC^®^ HSS T3 Chromatography Column (2.1 mm × 100 mm, 1.8 μm) at a column temperature of 40 °C. The mobile phases included 0.1% formic acid in water (A) and 0.1% formic acid in acetonitrile (B). The gradient elution procedure consisted of 2–5% B (0–0.3 min); 5% B (0.3–1.0 min); 5–20% B (1.0–7.0 min); 20% B (7.0–9.0 min); 20–25% B (9.0–9.5 min); 25–28% B (9.5–12.5 min); 28–40% B (12.5–18.0 min); 40–80% B (18.0–18.3 min); 80–98% B (18.3–21.0 min); 98% B (21.0–24.0 min). The mobile phase flow rate was 0.5 mL/min, and the injection volume was 1 μL.

The mass spectrometry acquisition was conducted in negative ion mode with an acquisition mode of MS*^E^*. The parameters were set as follows: capillary voltage, 2000 V; cone voltage, 40 V; desolvent gas, nitrogen; gas flow rate, 900 L/h; desolvation gas temperature, 450 °C; ion source temperature, 100 °C; mass range, *m*/*z* 50–1200; collision gas, argon gas; low energy scanning trap voltage, 6 eV; high energy scanning trap voltage, 50–70 eV. To correct the mass, Leucine enkephalin was used as the mass correction solution.

### 3.5. Quantitative Analysis of the Main Active Compounds by UPLC-MS/MS

The method used for quantitative analysis was based on Wang et al. [51] and optimized for this study. The quantitative analysis was performed on a UPLC (ACQUITY UPLC I-Class, Waters, Milford, MA, USA) tandem triquadrupole-mass spectrometer (QTRAP 6500, AB Sciex, Framingham, MA, USA) system. The chromatography column employed was the Waters Acquity UPLC^®^ BEH C18 Column (2.1 mm × 100 mm, 1.7 μm) with a temperature of 40 °C. The mobile phases used were 0.1% formic acid in acetonitrile (A) and 0.1% formic acid in water (B). The gradient elution procedure was set to 5–15% A (0–2.5 min); 15–35% A (2.5–6.0 min); 35–60% A (6.0–7.5 min); 60–95% A (7.5–8.0 min); 95% A (8.0–9.5 min); 95–5% A (9.5–10 min); 5% A (10–12 min). The flow rate of the mobile phase was 0.5 mL/min, and the injection volume was 1 μL.

Mass spectrometry acquisition was performed in negative ion mode using an electrospray ion source (EIS) with an ion source temperature of 550 °C. Quantitative analysis was performed using a Multi-response monitoring (MRM) model, and the mass spectrometry conditions, and their optimized condition parameters are provided in Table A1.

### 3.6. Microbial Strains and Culture Media

The tested microbial strains included *Staphylococcus aureus*, *Staphylococcus epidermidis*, *Bacillus subtilis*, *Bacillus cereus*, *Pseudomonas aeruginosa*, *Escherichia coli*, *Klebsiella aerogenes*, *Klebsiella pneumoniae*, and *Candida albicans*, all of which were purchased from BeNa Culture Collection (Beijing, China). Bacterial strains were cultured in Luria-Bertani medium at 37 °C after activation, whereas the fungal strains were cultured in Sabouraud medium at 30 °C. Single colonies from the plate were cultured in a liquid medium, and the turbidity of the bacterial suspension was diluted to 0.5 McFarland when the concentration of the bacterial suspension was 10^6^ cfu/mL.

### 3.7. Determination of Minimum Inhibitory Concentration (MIC) and Minimum Bactericidal Concentration (MBC)

The optimized broth microdilution method and 2,3,5-triphenyltetrazolium chloride (TTC) staining was used to determine the MIC and MBC of the methanol extracts and their main active compounds [11,52]. The alcohol extract (100 mg/mL) and 14 active compounds (20 mg/mL) were dissolved in dimethyl sulfoxide (DMSO) to prepare the stock solutions, which were then added to a 96-well microplate and diluted with the medium in the range of 0.156–5.00 mg/mL using the two-fold serial dilution method. The medium was inoculated with 0.5 × 10^6^ cfu/mL of the strain. The positive control was tetracycline or fluconazole (stock solutions concentration, 2 mg/mL) co-cultured with the bacterial suspension, while the negative control was the sample solvent co-cultured with the bacterial suspension. All wells contained no more than 2.5% DMSO and had a final volume of 200 μL. The MIC was determined as the lowest concentration at which the microorganisms did not demonstrate any visible growth after 24 h of incubation at 37 °C or 30 °C, and the MBC was defined as the lowest concentration at which the medium with 20 μL of 0.5% TTC solution added did not turn red.

### 3.8. Acute Toxicity Study

The mice were acclimatized and fed for 7 days prior to the experiments. Before administration, mice were fasted for 4 h, although they were allowed to drink freely. The MEAA dissolving in a 0.5% CMC-Na solution and administered as a single dose by gavage at 5000 mg/kg bw to one male and one female mouse. During the subsequent 24 h period, the activity level, hair condition, defecation, and death of these two mice were observed. Under normal conditions for the initial two mice, four male and four female mice were administered the same dose, totaling five treated mice in each group. Simultaneously, five male and five female mice were given 0.5% CMC-Na solution by gavage to establish the control group. The animals were observed individually for the first 30 min after treatment, mainly to observe the level of activity, hair condition, defecation, and general states. Special attention was given during the first 4 h, with regular observations during the first 24 h and daily thereafter for a total of 14 days. During this period, the body weight of each group of mice was measured and recorded daily. At the end of the observation period, the mice were weighed, humanely killed, and subjected to gross necropsy. The lesions of the major organs, including the heart, spleen, kidney, lung, liver, and thymus, were observed and weighed for the calculation of organ coefficients [51]. The data were analyzed using GraphPad Prism software (version 7.0, San Diego, CA, USA) and are presented as mean ± standard error of the mean (SEM). Significant differences among groups were assessed using one-way ANOVA. Statistical significance was determined at *p* < 0.05.

## 4. Conclusions

In this study, we evaluated the antimicrobial activity of MEAA, which has received limited attention in previous years. The extracts exhibited broad-spectrum antibacterial and antifungal activity. To understand the active compounds responsible for this activity, we summarized previous identification strategies and analyzed the phenolic compounds and their isomers in *A. absinthium*. We identified a total of 90 compounds, including flavonoids, quinic acids, and glucaric acids, with most of them being reported for the first time in *A. absinthium*. Notably, we identified a new class of caffeoyl methyl glucaric acids. We quantitatively analyzed and screened 14 primary active compounds for antimicrobial activity. Salicylic acid, caffeic acid, casticin, and 3,4-dicaffeoylquinic acid showed promising antimicrobial activity. In the acute toxicity study, the 5000 mg/kg bw dose of MEAA had no significant effect on the general status, body weight, and organ condition of the mice. This indicates that MEAA has a good safety profile. Our findings suggest that *A. absinthium* can be used as an antibiotic alternative to combat bacterial resistance, and the identification and screening of phenolic compounds provide a basis for further exploration of its antimicrobial properties.

## Figures and Tables

**Figure 1 ijms-24-12044-f001:**
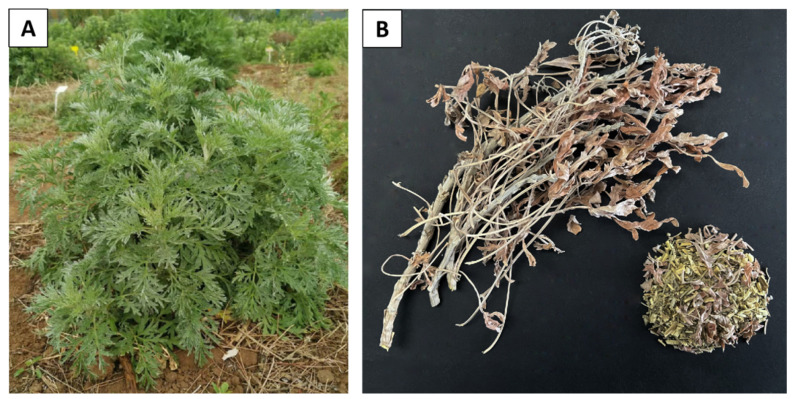
*A. absinthium* and its dried herbs. (**A**) *A. absinthium* in the wild state. (**B**) The dried herbs of *A. absinthium*.

**Figure 2 ijms-24-12044-f002:**
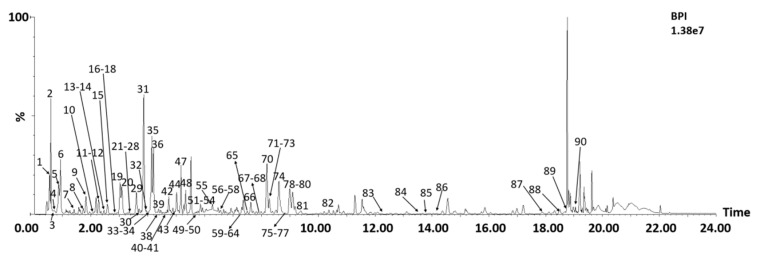
Base peak chromatograms of herbal extract of *A. absinthium* using UPLC-Q-TOF-MS. Compounds are numbered as listed in Table 1.

**Figure 3 ijms-24-12044-f003:**
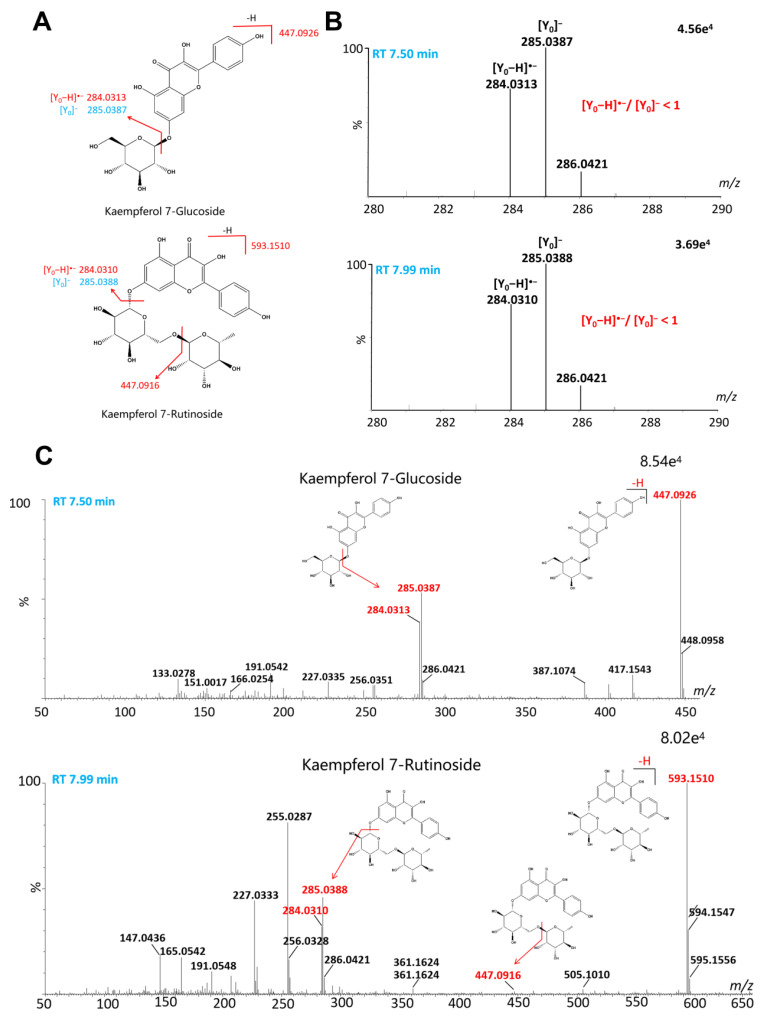
Identification of flavonoid 3,7-*O*-glycosides (taking kaempferol 7-glucoside and kaempferol 7-rutinoside, for example). (**A**) The structure and major fragmentation behavior of kaempferol 7-glucoside and kaempferol 7-rutinoside. (**B**) The major fragment ions [Y_0_-H]^−^ and [Y_0_]^−^ of them. (**C**) The MS/MS spectrum of them.

**Figure 4 ijms-24-12044-f004:**
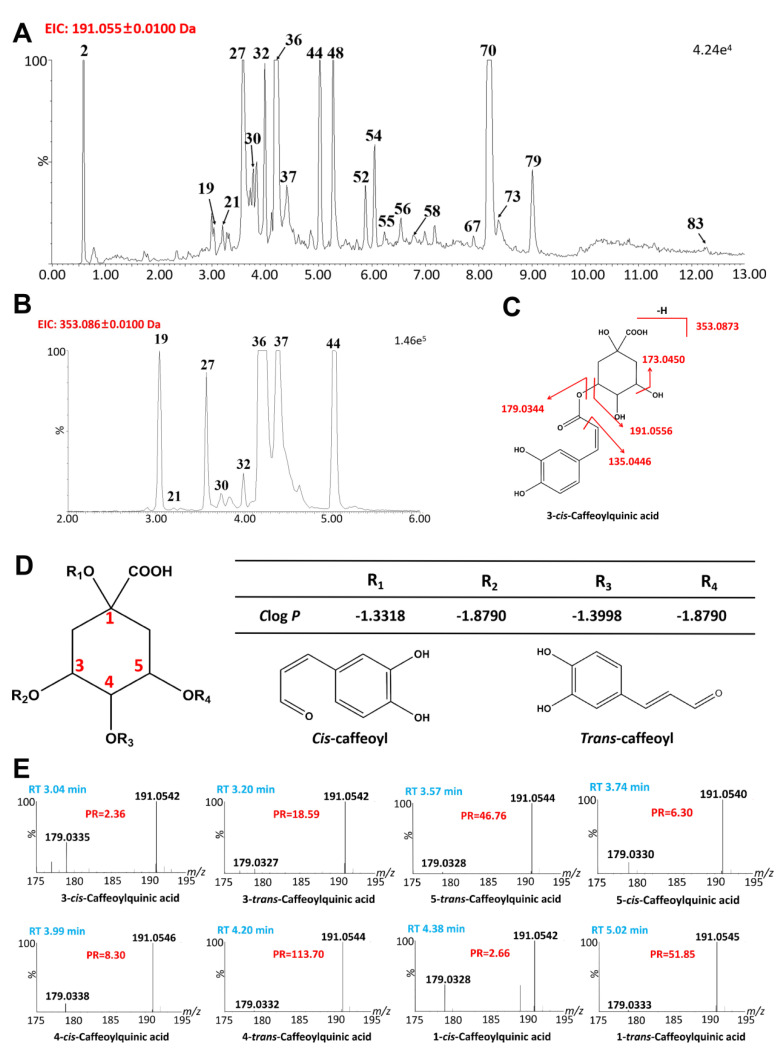
Identification of positional and geometric isomers of quinic acids using the Clog*P*s and the PRs. (**A**) Screening of QAs using the extraction MS/MS chromatogram (2-EIC). (**B**) The extraction MS chromatogram (1-EIC) of caffeoylquinic acid isomers. (**C**) The structure and major fragmentation behavior of caffeoylquinic acid (taking 3-*cis*-caffeoylquinic acid, for example). (**D**) The Clog*P* of caffeoylquinic acid isomers. (**E**) The PR of the DPI at *m*/*z* 191.06 to *m*/*z* 179.03 for eight isomers.

**Figure 5 ijms-24-12044-f005:**
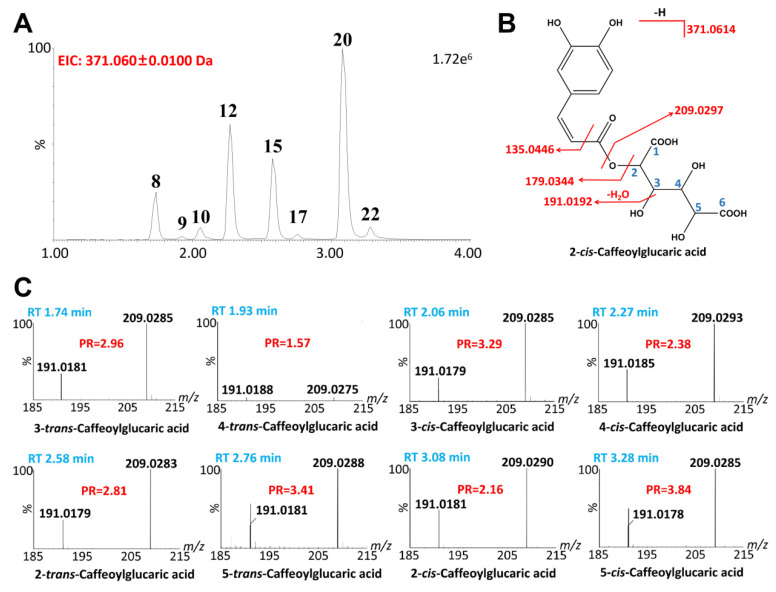
Identification of positional and geometric isomers of caffeoylglucaric acid. (**A**) The extraction MS chromatogram (1-EIC) of caffeoylglucaric acid isomers. (**B**) The structure and major fragmentation behavior of caffeoylglucaric acids (taking 2-*cis*-caffeoylglucaric acid, for example). (**C**) The PR of the DPI at *m*/*z* 209.03 to *m*/*z* 191.02 for eight isomers.

**Figure 6 ijms-24-12044-f006:**
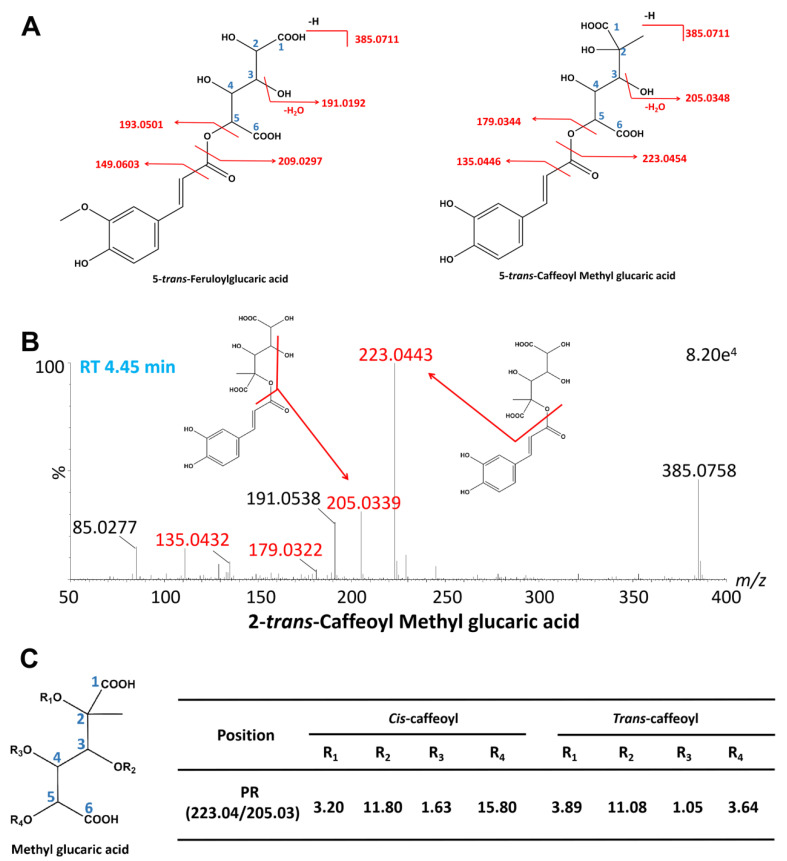
Identification of a new class of caffeoyl methyl glucaric acid isomers. (**A**) The structure and major fragmentation behavior of 5-*trans*-feruloylglucaric acid and 5-*trans*-caffeoyl methyl glucaric acid. (**B**) The MS/MS spectrum of 2-*trans*-caffeoyl methyl glucaric acid. (**C**) The PR of the DPI at *m*/*z* 223.04 to *m*/*z* 205.03 for eight isomers.

**Figure 7 ijms-24-12044-f007:**
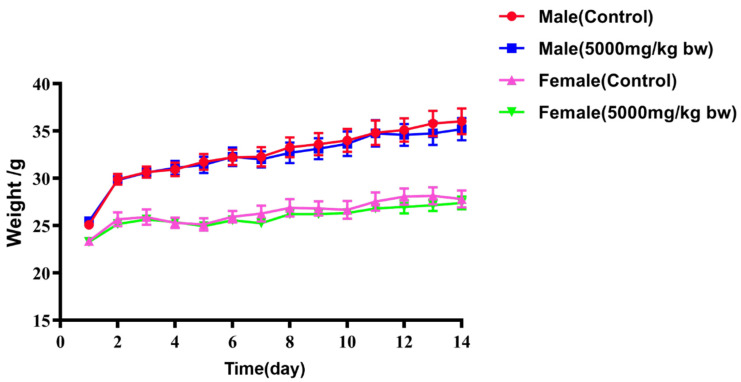
The body weight change of mice in each group for 14 days. Values expressed as mean ± SEM (*n* = 5).

**Table 1 ijms-24-12044-t001:** Antimicrobial activity of the methanol extracts of *A. absinthium*.

Microorganism		Methanol Extracts		Tetracycline		Fluconazole	
		MIC (mg/mL)	MBC (mg/mL)	MIC (μg/mL)	MBC (μg/mL)	MIC (μg/mL)	MBC (μg/mL)
G+ bacteria	*S. aureus*	1.25	1.25	<1.56	<1.56	NT	NT
	*S. epidermidis*	0.625	1.25	<1.56	<1.56	NT	NT
	*B. subtilis*	2.5	2.5	<1.56	<1.56	NT	NT
	*B. cereus*	1.25	1.25	12.5	12.5	NT	NT
G− bacteria	*P. aeruginosa*	1.25	1.25	12.5	25	NT	NT
	*E. coli*	2.5	2.5	3.125	3.125	NT	NT
	*K. aerogenes*	2.5	2.5	3.125	3.125	NT	NT
	*K. pneumoniae*	2.5	2.5	3.125	3.125	NT	NT
Fungal strains	*C. albicans*	2.5	5.0	NT	NT	5.0	10

NT = Not tested.

**Table 3 ijms-24-12044-t003:** Content of 14 active compounds in *A. absinthium* (*n* = 3).

Compound	Linear Formula	R^2^	Content/(μg/g)
Salicylic acid	y = 3694.3x + 283,353	0.9995	0.3601 ± 0.019
Caffeic acid	y = 5170.6x + 305,047	0.9996	1.8934 ± 0.032
Rosmarinic acid	y = 861.98x + 6146.5	1.0000	0.1515 ± 0.039
Chlorogenic acid	y = 1597.7x + 19,247	1.0000	25.5950 ± 0.544
Cryptochlorogenic acid	y = 2623.3x − 6771.2	1.0000	0.6693 ± 0.011
Neochlorogenic acid	y = 2968.9x − 67,665	0.9999	1.3589 ± 0.029
3,4-Dicaffeoylquinic acid	y = 2513.4x − 18,469	1.0000	0.9548 ± 0.055
4,5-Dicaffeoylquinic acid	y = 1952.4x − 41,101	0.9999	1.9678 ± 0.016
Kaempferol	y = 144.18x + 7766	0.9995	0.2022 ± 0.057
Apigenin	y = 852.56x + 54,793	0.9995	0.5389 ± 0.099
Casticin	y = 201.8x + 7446.7	0.9998	3.5470 ± 0.148
Rutin	y = 1265.4x − 7408.8	1.0000	11.3730 ± 0.368
Isoquercitrin	y = 2258.3x + 12,910	1.0000	0.7482 ± 0.024
Kaempferide	y = 728.34x + 11,583	1.0000	8.8500 ± 0.258

**Table 4 ijms-24-12044-t004:** Antimicrobial activity of 4 active compounds of *A. absinthium*.

Microorganism	Salicylic Acid		Caffeic Acid		Casticin		3,4-Dicaffeoylquinic Acid	
	MIC (mg/mL)	MBC (mg/mL)	MIC (mg/mL)	MBC (mg/mL)	MIC (mg/mL)	MBC (mg/mL)	MIC (mg/mL)	MBC (mg/mL)
*S.aureus*	0.25	0.5	0.5	0.5	0.5	0.5	0.5	0.5
*S. epidermidis*	0.25	0.5	0.5	1	1	1	1	>1
*B. subtilis*	0.5	0.5	1	1	0.5	1	0.5	1
*B. cereus*	0.5	0.5	0.5	1	1	1	1	>1
*P. aeruginosa*	0.25	0.5	0.5	1	1	1	0.5	1
*E. coli*	0.5	0.5	1	>1	1	>1	1	1
*K. aerogenes*	0.5	1	0.5	1	1	>1	>1	NT
*K. pneumoniae*	0.25	0.5	1	1	1	1	1	>1
*C. albicans*	0.5	1	1	>1	>1	NT	>1	NT

Note: NT = Not tested.

**Table 5 ijms-24-12044-t005:** Body weight of mice in each group on day 14.

Group	Weight (g)	
	Male	Female
Control	36.04 ± 1.35 ^ns^	27.82 ± 0.90 ^ns^
5000 mg/kg bw	35.20 ± 117 ^ns^	27.40 ± 0.66 ^ns^

Note: Values expressed as mean ± SEM (*n* = 5). “ns”: no significant differences.

**Table 6 ijms-24-12044-t006:** Organ coefficients of each group of mice.

Group		Heart	Liver	Spleen	Lung	Kidney	Thymus
Male	Control	5.62 ± 0.22 ^ns^	55.74 ± 1.06 ^ns^	3.96 ± 0.27 ^ns^	6.39 ± 0.31 ^ns^	16.34 ± 0.90 ^ns^	2.75 ± 0.32 ^ns^
	5000 mg/kg bw	5.66 ± 0.49 ^ns^	53.15 ± 0.40 ^ns^	4.21 ± 0.28 ^ns^	6.11 ± 0.40 ^ns^	17.03 ± 1.37 ^ns^	2.22 ± 0.26 ^ns^
Female	Control	6.66 ± 0.41 ^ns^	49.45 ± 0.46 ^ns^	5.06 ± 0.29 ^ns^	6.64 ± 0.34 ^ns^	12.31 ± 0.49 ^ns^	3.32 ± 0.31 ^ns^
	5000 mg/kg bw	5.75 ± 0.25 ^ns^	47.76 ± 0.61 ^ns^	4.58 ± 0.39 ^ns^	6.57 ± 0.23 ^ns^	12.63 ± 0.25 ^ns^	3.37 ± 0.20 ^ns^

Note: Values expressed as mean ± SEM (*n* = 5). “ns”: no significant differences.

## Data Availability

The data that support the findings of this study are available from the corresponding author upon reasonable request.

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
