# Peer review of "New Evidence for Artemisia absinthium as an Alternative to Classical Antibiotics: Chemical Analysis of Phenolic Compounds, Screening for Antimicrobial Activity"

_ijms, 2023, doi:10.3390/ijms241512044_

Round 1

Reviewer 1 Report

The manuscript is well written and logically presented. But my estimates would be higher if the authors synthesized at least one, or rather several, compounds from the new class of caffeoyl methyl glucaric acids and showed that the retention time of the real compound coincides with the expected one. The main remark to the work refers to the same topic. I recommend writing more carefully about the identification of new compounds: a structure is proposed, possible structures.

A few small notes:

1) Page 3, line 1: gluconic should be changed to glucaric?

2) Page 8. “They usually combine with aglycone to form flavonoid glycosides”. Flavonoids usually combine with sugar residue

3) Page 12, penultimate line: there are no acyl groups in the structure of glucaric acid.

4) The last paragraph of section 2.5: the text about the safety of other plant extracts should be removed. Its presence would make sense if we were talking about plants of the same genus. And so, I can name a large number of plant extracts, a drop of which can kill an animal.

Author Response

Thank you for your professional advice. Please see the attachment for detailed responses.

Reviewer 2 Report

The manuscript presents the results of research conducted on Artemisia absinthium. The authors determined the chemical composition of the ethanol extract obtained from this plant, identifying 90 compounds, which I consider a very great achievement. In addition, they tested the activity of both the extract and selected specific compounds against 8 strains of bacteria and one strain of fungus.

The manuscript is interesting, well written, and the research is well documented.

Minor remarks

I think the term "alcohol extracts" should be changed to methanol extracts, both in the Abstract and in the text of the manuscript.

On page 2, the sentence ends with the word "and "in the phrase "stomachic and tonic properties and." This should be corrected.

In section 2.1, the S. epidermidis strain is mentioned twice, in the sentence "S. epidermidis (MIC=0.625 mg/mL, MBC=1.25 mg/mL), S. epidermidis, and B. cereus (MIC=1.25 mg/mL, MBC=1.25 mg/mL)." The latter is incorrect and needs to be changed.

In Table 1, there is an unnecessary bottom edge under the information about the S. aureus strain. Also, I don't understand why the data for this strain is shown in bold (similarly G+ bacteria)?

Table 2 - "[M-H]- measured" and [2M-H]- for compounds 9 and 11 are formatted incorrectly. Where there are two items as References, one of the parentheses is on the next line, this should be corrected.

In Table 2, the names of compounds are capitalized; in the text, they are either capitalized or lowercase. This should be standardized.

On page 9 in the phrase "and we identified. O-glucosyl flavonoids in this study" there is an unnecessary period.

On page 17, the word higher "10 compounds had higher MBC values higher than" is repeated twice. This should be corrected.

Item 3.4, 3.5, 3.7 - instead of uL it should be μL.

Item 3.5 - "The gradient elution procedure was set to 5% A (0-2.5 min); 5-15% A (0-2.5 min)" the same time range is given twice, should it be so?

"95% A (8.0-9.5 min); 5% A (9.5-12 min)." Should the last one be 5% A?

Item 3.6 - all strain names must be in italics.

Author Response

(The authors gave the same response as above.)

Reviewer 3 Report

The manuscript entitled "New Evidence for Artemisia absinthium as an Alternative to Traditional Antibiotics: Chemical Analysis of Phenolic Compounds, Screening for Antimicrobial Activity authored by Zhihao Liu et al presents the results obtained using both in vitro and in vivo studies on the chemical composition and biological properties of Artemisia absinthium 

Investigations of natural products as sources of compounds with therapeutic properties are very important and highly justified, but the study has little novelty. The herbal species Artemisia absinthium antimicrobial efficacy is well documented by the literature, along with other therapeutical properties. Furthermore, the antimicrobial properties are investigated only for MIC and MBCs against reference strains 

Did the authors consider the use of in vitro studies to evaluate the toxicity ? 

The discussion sections are not acceptable, for example:

"Similar to some other Artemisia species such as A. annua, A. argyi and A. indica, A. absinthium also exhibited antimicrobial activity, particularly against Gram-positive bacteria. This may be due to differences in the structure of cell membranes between Grampositive and Gram-negative bacteria. Hydrophilic compounds, which are present in the alcohol extracts, require permeation across the membrane to be effective. However, the presence of lipopolysaccharides in the outer membrane of Gram-negative bacteria can limit the penetration of hydrophobic antibacterial compounds, resulting in the need for higher doses of these compounds [25,26]. Further research is needed to determine the chemical profile and content of the compounds in the alcohol extracts, as well as the sensitivity of the tested strains."

The authors do not acknowledge their results and do not compare them with previous studies, while the literature contains a multitude of studies on the chemical and antimicrobial patterns of the herbal species.

As for the identified compounds, these are presented as important for antimicrobial efficacy, but again, no novelty and vague prospects 

"The screening of 14 compounds for antimicrobial activity in this study may not provide a precise indication of the potential antimicrobial effect of AEAA, which supports the multi-component, multi-targeted action of traditional herbal medicine in the treatment of diseases [49]. Further trials are still required to validate these potential compound synergies. Nonetheless, the screening results provide some evidence for the antimicrobial effects of A. absinthium."

The conclusions are not supported by the results and the discussion, the authors are recommended to be objective and accurate. 

"Conclusion In this study, we evaluated the antimicrobial activity of AEAA, which has received limited attention in previous. The extracts exhibited broad-spectrum antibacterial and antifungal activity. To understand the active compounds responsible for this activity, we summarized previous identification strategies and analyzed the phenolic compounds and their isomers in A. absinthium. We identified a total of 90 compounds, including flavonoids, quinic acids, and glucaric acids, with most of them being reported for the first time in A. absinthium. Notably, we identified a new class of caffeoyl methyl glucaric acids. We quantitatively analyzed and screened 14 primary active compounds for antimicrobial activity. Salicylic acid, caffeic acid, casticin and 3,4-dicaffeoylquinic acid showed promising antimicrobial activity. In the acute toxicity study, the 5000 mg/kg bw dose of AEAA had no significant effect on the general status, body weight and organ condition of the mice. This indicates that AEAA has a good safety profile. Our findings suggest that A. absinthium can be used as an antibiotic alternative to combat bacterial resistance, and the identification and screening of phenolic compounds provide a basis for further exploration of its antimicrobial properties."

Along with these major flaws of the manuscript, the following aspects should be considered by the authors:

1. corrections for several statements of the text in case of methodology, results presentation

- correction for "traditional" antibiotics - classical antibiotics

- adding the type of extract for the corresponding phrases - alcoholic is not sufficient, the type of solvent is known as a factor related to variation in chemical composition and biological activities

- the authors refer to extracts, but the results are presented for one extract, to 14 active compounds, but the results are presented for 4 compounds

- organism name (Latine style), reference strain code

-

Moderate editing of English language required

Author Response

(The authors gave the same response as above.)
